# A Direct Three-Phase AC–AC Matrix Converter-Based Wireless Power Transfer System for Electric Vehicles

**Rutian Wang [1],\*** , **Min Huang [1]**, **Chongyi Lu [2]** and **Weiquan Wang [2]**

[1]  Key Laboratory of Modern Power System Simulation and Control & Renewable Energy Technology, Ministry of Education (Northeast Electric Power University), Jilin 132012, China; nimgnauh@gmail.com
[2]  State Grid Liaoning Power Co., Ltd. Dandong Power Supply Company, Dandong 118000, China; rogerziv@163.com (C.L.); oliver08l@163.com (W.W.)
\*  Correspondence: wrtmail@163.com; Tel.: +86-159-4869-6698

**Abstract:** For the bidirectional wireless power transfer system of electric vehicles, the topology proposed in this paper includes a direct three-phase AC–AC matrix converter as the pre-stage main circuit, a bilateral inductor–capacitor–capacitor–inductor (LCCL) as the resonance compensation network, and a full-bridge converter as the latter stage circuit. According to the characteristics of the system topology, a new control strategy is proposed based on the de-re-coupling method. The control principle of the coordination work of the scheme is expounded, and the corresponding switch combination logic is designed. According to the three-phase voltage amplitude relationship at different stages, combined with the resonant frequency, the switch arms of the matrix converter are alternately controlled separately. The number of switching operations is reduced, the system efficiency and safety are improved, and the full range of soft switching operations of the converter is realized. The theoretical analysis of the bilateral LCCL resonance compensation network is carried out, and its constant voltage/current output characteristic and high power factor transmission characteristic are obtained. Finally, the effectiveness and feasibility of the bidirectional wireless power transfer system for electric vehicles proposed in this paper are verified by simulation analysis.

**Keywords:** AC–AC converter; bidirectional wireless power transfer; soft switching; bilateral LCCL compensation

## 1. Introduction

With the exhaustion of energy and the growing environmental problems, the demand for energy-saving and environmentally-friendly new vehicles has increased significantly. Electric vehicles are the products that emerged under this trend. The emergence of the electric vehicles has made great contributions to promoting the development of modern society and the progress of civilization. The power used for electric vehicles is rich in resources, easy to transmit, and environmentally friendly, which can effectively cope with energy and environmental problems [1]. The charging methods of electric vehicles are divided into wired charging and wireless charging. In most cases, the charging mode of electric vehicles is wired charging. However, in this way, there are safety hazards such as bare wires and plugs in the charging process. In some special circumstances, such as chemical exposure, mining, and other inflammable and explosive situations, the resulting electric spark is enough to cause a huge safety accident. In recent years, wireless charging, as a technology for transmitting power, could be done without any physical contact, and it has attracted people's attention. Compared to traditional wired power transmission, it has many advantages, such as high current isolation, high flexibility, and high reliability [2].

Inductive power transfer (IPT) technology is used for wireless power transfer of electric vehicles. In a loosely coupled IPT system, the inductive link needs to generate a strong magnetic field so that energy can be transmitted from the outside of the vehicle to the vehicle receiving end through the air gap [3]; this requires the power frequency AC to be raised to high-frequency AC. The conversion of electric energy necessarily requires the addition of power electronic converters. At present, the most common converter structure is a two-stage conversion structure; that is, an AC–DC–AC type structure, which generally involves a low-frequency ac-to-dc power conversion as well as a dc-to-high-frequency ac power conversion, requiring two or more power electronic converters in conjunction with a large and expensive dc-link capacitor and a bulky low-frequency input inductor [4,5].

In order to solve the above situation, matrix converters have been used in recent years to replace two-stage converters for inductive power transfer (IPT) systems. As a direct AC–AC converter, the matrix converter has a simple and compact topology, bidirectional power flow capability, a high quality input current waveform, and an adjustable input power factor compared with conventional converters [6–11]. In [12], a single-phase matrix converter is used to alternately control the opening and closing of the switching tube by a phase shift of a certain angle. The method is simple. However, the switch cannot be turned on or off at zero voltage or zero current, and the switching loss is large. The study in [13] used a three-phase matrix converter, which can only realize the soft-switching start of the switching tube. When the circuit is turned off, the current is not zero, and the switching loss is large. The space vector pulse width modulation (SVPWM) control strategy is used in [14,15] to make the converter have good input and output characteristics, but the strategy control is more complicated and cannot achieve full range of soft switching operation. In [16], based on energy injection and free oscillation technology, the soft switching operation and high efficiency of the converter can be realized. However, this method requires precise control of the energy injection time and the direction of the detected current, the control is complicated, and the converter has only the ability to flow in one direction.

The IPT system needs to reduce the volt-ampere capacity and improve the power transmission capability through the resonance compensation circuit [17,18]. According to the difference between the resonant capacitor and the coil connection, it can be divided into four basic compensation circuits; namely, series-series (SS), series-parallel (SP), parallel-series (PS), and parallel-parallel (PP) [19–21]. It is stated in the literature [17,19–23] that these traditional methods are affected to varying degrees by the load and coil coupling coefficients. In the SS structure, the resonant capacitance of the transmitting end does not change with the load or the coupling ratio. In the SP structure, the resonant capacitance of the transmitting end changes with the coupling rate. In the PS and PP structures, the resonant capacitance of the transmitting end changes with the load and the coupling ratio. In the IPT system of the [24,25], the inductor–capacitor–inductor (LCL) resonance compensation circuit is adopted at the transmitting end. At the resonant frequency, the transmitting end coil exhibits a constant current characteristic, which does not change with the change of the load and the coupling ratio, and the transmitting end always works at the maximum excitation state; the receiving end uses a parallel resonant network. However, when the receiving end adopts the parallel resonant network, the coil contains reactive components, and the reactive component changes with the load and the lateral and longitudinal offset, and is fed back to the transmitting end through mutual inductance. For this reason [26] proposed a bilateral LCL type compensation circuit to solve the shortcomings caused by the traditional compensation method, so that the system has constant voltage constant current output characteristics. However, its compensation inductance is large and uneconomical, and the component loss is large.

In view of the above problems, this paper proposes a new type of electric vehicle wireless charging and discharging system based on direct three-phase AC–AC matrix converter, which can realize the conversion of low-frequency alternating voltage to high-frequency alternating voltage without any intermediate energy storage components. At the same time, for the traditional capacitor-inductor series/parallel compensation affected by the load and coil coupling coefficient, the structure of the

bilateral LCCL resonant network is proposed. Combined with the above system structure, a new control strategy is proposed. The idea of de-re-coupling is adopted to reduce the number of switching times and make the switch work in the soft switching state at the same time, reducing switching loss and improving system efficiency, thereby achieving the purpose of charging and discharging electric vehicles.

The structure of this paper is as follows: the second section introduces the topology of the proposed system; the third section introduces the control strategy adopted by the proposed system; the fourth section theoretically analyzes the bilateral LCCL resonant networks; the fifth section gives simulation analysis; and finally, the sixth section gives the conclusion.

## 2. System Topology

The IPT system based on direct three-phase AC–AC matrix converter proposed in this paper is shown in Figure 1. The primary side consists of a three-phase AC–AC matrix converters and an LCCL resonant tank, and the secondary side consists of a single-phase full-bridge and an LCCL resonant tank. The proposed topology eliminates input inductor, DC link capacitor, and AC–DC–AC two-stage power conversion. Therefore, it achieves a direct conversion between low-frequency AC and high-frequency AC. The matrix converter consists of six bidirectional switches ($S_1$–$S_6$), each consisting of two back-to-back connected IGBT switches. In Figure 1, $L_1$ and $L_2$ respectively indicate the self-inductance of the primary side coil of the magnetic coupler and the self-inductance of the secondary side coil, and $M$ is the mutual inductance between the primary and the secondary coils. The compensation network consists of a magnetic coupler and three other components: the series inductor $L_s$ and $L_p$, the shunt capacitor $C_s$ and $C_p$, and the series capacitor $C_1$ and $C_2$. In addition, $v_s$ represents the voltage applied by the inverter to the input end of the resonant network, and $v_p$ represents the voltage at the output end of the secondary side compensation network. $i_1$, $i_2$, $i_s$, and $i_p$ represent the currents flowing through the inductors $L_1$, $L_2$, $L_s$, and $L_p$ respectively.

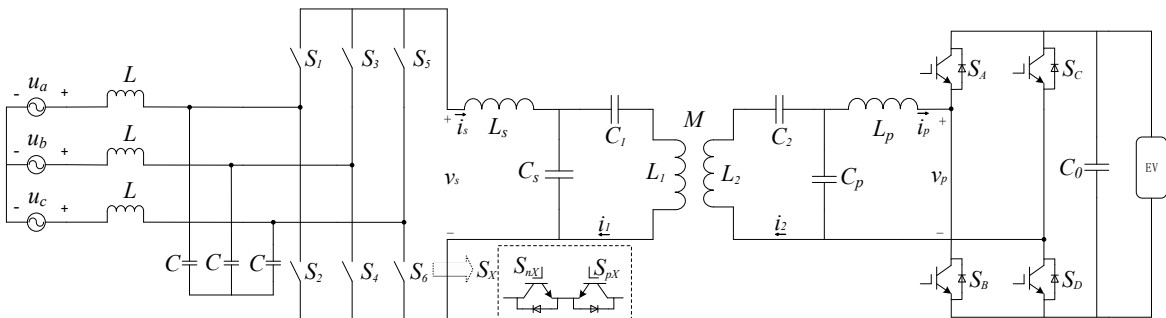

**Figure 1.** Electric vehicle wireless charging system based on direct three-phase AC–AC matrix converter.

## 3. Control Strategy

### 3.1. Matrix Converter

According to the circuit topology characteristics shown in Figure 1, the matrix converter can be converted into common converters for analysis by de-re-coupling. Therefore, modulation and control strategies widely used in conventional single-phase switching converters can be applied to matrix converters, making the analysis and control of bidirectional switching circuits simpler. In this paper, based on the de-re-coupling method, a de-re-coupling vector modulation strategy suitable for direct three-phase AC–AC matrix converter is proposed. The proposed strategy also needs to be processed by "two-step" de-coupling and re-coupling. Firstly, direct three-phase AC–AC matrix converter is processed by the "de-coupling" idea, and the original topology is presented as a form of a positive six-switch converter and a reverse six-switch converter as shown in Figure 2. When the positive converter is operated, the forward current flows through the anti-parallel diodes of the reverse

converter switches, and the switch tubes corresponding to the reverse converter are all turned off. When the reverse converter is operated, the reverse current flows through the anti-parallel diodes of the positive converter switches, and the switch tubes corresponding to the positive converter are all turned off.

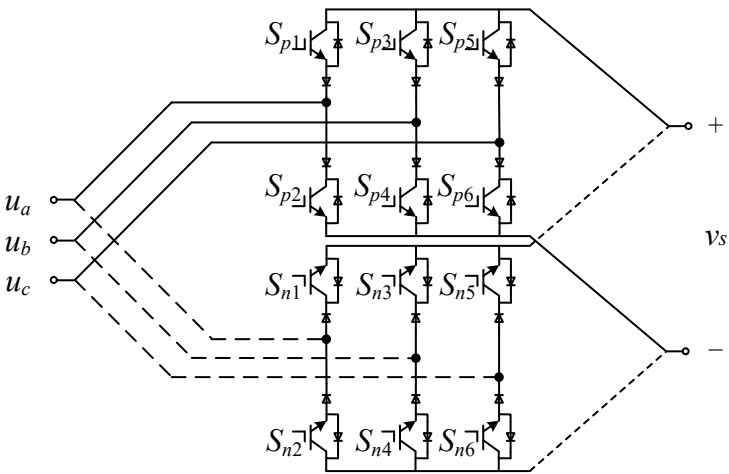

**Figure 2.** De-coupling diagram of direct three-phase AC–AC matrix converter.

Secondly, the "re-coupling" work focuses on generating drive pulses for each switching device in the "de-coupling" circuit through logical combination and transformation. The research object of this paper is the voltage type matrix converter. For this reason, the control mode of Figure 3 is selected in the driving mode of the bidirectional switch (taking a single bridge arm as an example); that is, two switching tubes are respectively applied with different driving, and are separately controlled. $S_{amax}$ and $S_{amin}$ are the driving signals of the A-phase arm; and the polarity selection signal $V_u$ and the polarity selection signal $V_n$ are complementary high-frequency square waves with a duty ratio of 50%, and the frequency is the same as the switching frequency. In the first half of the polarity selection's high-frequency square wave signal, the matrix converter outputs a positive voltage, which is equivalent to turning off the reverse converter and driving the positive converter. In the latter half of the period, when the polarity selects the high-frequency square wave signal, the matrix converter outputs a negative voltage, which is equivalent to turning off the positive converter and driving the reverse converter. Thereby, the matrix converter output can be realized as a high-frequency alternating voltage in one high-frequency period.

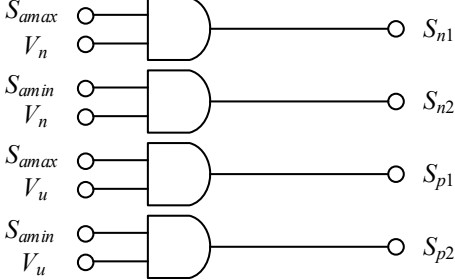

**Figure 3.** The re-coupling logic of the A phase drive signal.

In Figure 4, taking the ab phase voltage as an example, the output voltage $v_s$ of the matrix converter is synthesized by the polarity selection signals $v_u$, $v_n$.

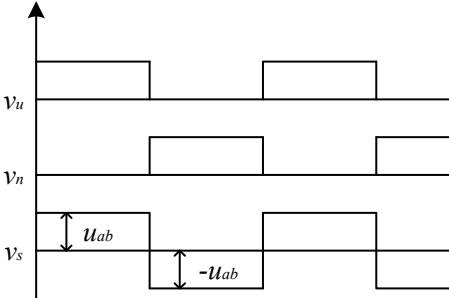

**Figure 4.** Synthesis of $v_s$.

As shown in Table 1, the three-phase AC–AC matrix converter has twelve operating states throughout the cycle, which are determined based on the polarity selection signal and the relative order of the three-phase input voltages.

$$S_{n1} = S_{ab} \cdot \overline{S_{ca}} \cdot (\overline{S_p} \cdot S_f + S_p \cdot S_v)$$
$$S_{p1} = \overline{S_{ab}} \cdot S_{ca} \cdot (S_P \cdot S_f + \overline{S_p} \cdot S_v) \tag{1}$$

**Table 1.** Twelve operating states of the matrix converter.

| Mode | $S_u$ | $S_{ab}$ | $S_{bc}$ | $S_{ca}$ | Switches | |
|------|-------|----------|----------|----------|----------|---------|
| | | | | | Forward | Reverse |
| 1 | 0 | 1 | 1 | 0 | $S_{n1}, S_{n6}$ | $S_{p2}, S_{p5}$ |
| 2 | 0 | 1 | 0 | 0 | $S_{n1}, S_{n4}$ | $S_{p2}, S_{p3}$ |
| 3 | 0 | 0 | 1 | 0 | $S_{n3}, S_{n6}$ | $S_{p4}, S_{p5}$ |
| 4 | 0 | 0 | 1 | 1 | $S_{n3}, S_{n2}$ | $S_{p4}, S_{p1}$ |
| 5 | 0 | 0 | 0 | 1 | $S_{n5,}, S_{n2}$ | $S_{p6}, S_{p1}$ |
| 6 | 0 | 1 | 0 | 1 | $S_{n5}, S_{n4}$ | $S_{p6}, S_{p3}$ |
| 7 | 1 | 1 | 1 | 0 | $S_{p2}, S_{p5}$ | $S_{n1}, S_{n6}$ |
| 8 | 1 | 1 | 0 | 0 | $S_{p2}, S_{p3}$ | $S_{n1}, S_{n4}$ |
| 9 | 1 | 0 | 1 | 0 | $S_{p4}, S_{p5}$ | $S_{n3}, S_{n6}$ |
| 10 | 1 | 0 | 1 | 1 | $S_{p4}, S_{p1}$ | $S_{n3}, S_{n2}$ |
| 11 | 1 | 0 | 0 | 1 | $S_{p6}, S_{p1}$ | $S_{n5}, S_{n2}$ |
| 12 | 1 | 1 | 0 | 1 | $S_{p6}, S_{p3}$ | $S_{n5}, S_{n4}$ |

The logic signal of switch $S_1$ of the converter can be obtained from Table 1 as shown in Equation (1), where the variables are defined as $S_{ab}$: $u_a > u_b$, $S_{bc}$: $u_b > u_c$, $S_{ca}$: $u_c > u_a$, $S_u$: $V_u > 0$, $S_f$: forward power transfer mode, $S_v$: reverse power transfer mode. The logic signals of other switches ($S_2$–$S_6$) are similar to Equation (1).

### 3.2. Full-Bridge Converter

In Figure 1, the full-bridge converter is connected to the resonant tank and the electric vehicle. Depending on the direction of the power flow, the electric vehicle can be used as a power source or as an active load. The operation of the full-bridge converter is defined based on the phase modulation principle similar to that of MC. Hence, the switches $S_A$–$S_D$ of the converter are operated at $f_T$ with 50% duty cycle to produce $v_p$.

In Table 2, the full-bridge converter has two operational states throughout the cycle, and these two operational states are determined based on the power flow direction.

**Table 2.** Two operating states of the full-bridge converter.

| Mode | $v_p > 0$ | Switches | |
| --- | --- | --- | --- |
| | | Forward | Reverse |
| 1 | 1 | $S_A, S_D$ | $S_B, S_C$ |
| 2 | 0 | $S_B, S_C$ | $S_A, S_D$ |

Figure 5 shows the $v_s$ and $v_p$. The transmission power can be changed by changing θ.

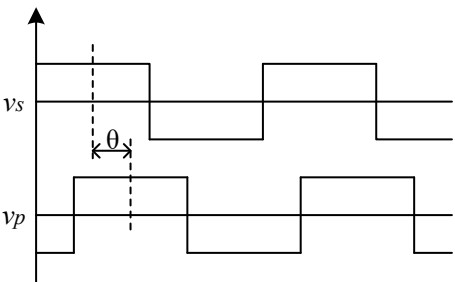

**Figure 5.** $v_s$ and $v_p$.

## 4. Analysis of Bilateral LCCL Resonant Networks

This chapter uses a bilateral LCCL compensation method. The IPT resonant circuit topology using this compensation method is shown in Figure 1.

For such a high-order system, in order to facilitate the analysis of basic characteristics, the following two approximations need to be made: (1) ignoring the internal resistance of all passive components in the circuit; (2) using the fundamental analysis method—i.e., assuming that only the fundamental component in square wave $v_s$ and $v_p$ plays a major role in energy transmission. Thus, the bilateral LCCL-compensated IPT resonant network equivalent circuit of Figure 1 can be represented as shown in Figure 6. The physical quantity with the superscript in the figure indicates the derivation value of the secondary side to the primary side.

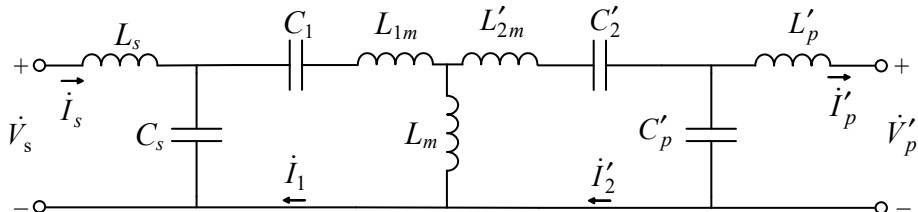

**Figure 6.** The re-coupling logic of the A phase drive signal.

In this circuit topology, the ratio of the magnetic coupler is defined as $n = L_1/L_2$. The other physical quantities in Figure 6 are defined in Equation (2).

$$
\begin{aligned}
L_m &= k \cdot L_1 \\
L_{1m} &= (1-k) \cdot L_1 \\
L'_{2m} &= (1-k) \cdot nL_2 \\
C'_2 &= C_2/n \\
C'_p &= C_p/n \\
L'_p &= n \cdot L_p \\
v'_p &= \sqrt{n} \cdot v_p
\end{aligned}
\tag{2}
$$

In Equation (2), $k$ represents the coupling coefficient of the primary and secondary coils.

Although a high-order system usually has multiple resonant frequencies, the original intent of this chapter is to find a compensation topology suitable for fixed-frequency control to simplify the design of the control system. Therefore, the characteristics of the proposed topology over the entire frequency domain will not be discussed; there will be an analysis of the characteristics of the resonant frequency point that satisfies Equation (3) only.

$$\omega_0^{-2} = L_s C_s = L_p C_p = (L_1 - L_s)C_1 = (L_2 - L_p)C_2 \tag{3}$$

In Equation (3), $\omega_0$ is the resonant angular frequency.

For ease of analysis, the series of $C_1$ and $L_{1m}$ in Figure 6 can be represented by an equivalent inductance, which is Equation (4).

$$L_{1s} = \frac{1}{j\omega_0} \cdot \left( \frac{1}{j\omega_0 C_1} + j\omega_0 L_{1m} \right) = L_{1m} - \frac{1}{\omega_0^2 C_1} \tag{4}$$

Substituting the corresponding expressions of Equations (2) and (3) into Equation (4) is (5).

$$L_{1s} = L_1 - \frac{1}{\omega_0^2 C_1} - kL_1 = L_s - kL_1 \tag{5}$$

Similarly, $C'_2$ and $L'_{2m}$ can also be equivalent to Equation (6).

$$L'_{2p} = L'_p - kL_1 \tag{6}$$

After the above equivalent, according to the superposition theorem, the effects of the input voltage $\dot{V}_s$ and the output voltage $\dot{V}'_p$ on the resonant network can be separately analyzed. First, consider the case where only the input terminal voltage $\dot{V}_s$ is applied to the resonant network, as shown in Figure 7; the broken line in the figure indicates that no current flows through the branch.

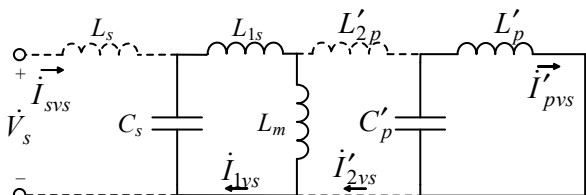

**Figure 7.** Circuit state of a resonant network acted on by $\dot{V}_s$ only.

From Equations (2) and (5), we can get Equation (7).

$$L_{1s} + L_m = L_s - k \cdot L_1 + k \cdot L_1 = L_s \tag{7}$$

Obviously, Equation (7) means that the second cell from the left in Figure 5 can also form a resonance. Then, under the action of $\dot{V}_s$, since $C_s$ and $(L_{1s} + L_m)$, $L'_p$ and $C'_p$ form a stable parallel resonance state at the resonant frequency, Equation (8) is inevitably caused.

$$\dot{I}_{svs} = 0, \quad \dot{I}'_{2vs} = 0 \tag{8}$$

The subscript vs. indicates that the current is generated only by the action of $\dot{V}_s$.

The establishment of Equation (8) means

$$\dot{V}_s = \dot{V}_{Csvs}, \quad \dot{V}_{Lmvs} = \dot{V}'_{Cpvs} \tag{9}$$

Further, we can easily get

$$\dot{I}_{1vs} = \frac{\dot{V}_s}{j\omega_0 L_s} \tag{10}$$

$$\dot{I}'_{pvs} = \frac{k\dot{V}_s L_1}{j\omega_0 L_s L'_p} \tag{11}$$

Similarly, the second cell from the right in Figure 8 can also form resonance. Then, when only $\dot{V}'_p$ acts, the current phasors of the resonant network can also be obtained in the same way.

$$
\begin{aligned}
\dot{I}_{1vp} &= 0, \quad \dot{I}'_{pvp} = 0 \\
\dot{I}'_{2vp} &= -\frac{\dot{V}'_p}{j\omega_0 L'_p} \\
\dot{I}_{svp} &= -\frac{k\dot{V}'_p L_1}{j\omega_0 L_s L'_p}
\end{aligned}
\tag{12}
$$

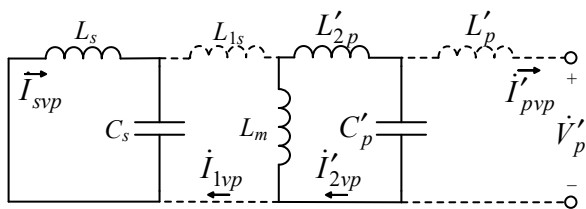

**Figure 8.** Circuit state of a resonant network acted on by $\dot{V}'_p$ only.

We assume that the relative phase angle between the voltages vs. and $v_p$ is $\theta$. If vs. is used as a reference voltage, then the two voltage phasors can be expressed as

$$
\begin{aligned}
\dot{V}_s &= V_s \angle 0° \\
\dot{V}'_p &= V'_p \angle \theta - 90°
\end{aligned}
\tag{13}
$$

Substituting Equation (2) into Equations (10) to (12), we can get the current phasors in the resonant network.

$$\dot{I}_s = \frac{k\sqrt{L_1 L_2} v_p}{\omega_0 L_s L_p} \angle \theta \tag{14}$$

$$\dot{I}_1 = \frac{v_s}{\omega_0 L_s} \angle -90° \tag{15}$$

$$\dot{I}_2 = \frac{v_p}{\omega_0 L_p} \angle \theta \tag{16}$$

$$\dot{I}_p = \frac{k\sqrt{L_1 L_2} v_s}{\omega_0 L_s L_p} \angle -90° \tag{17}$$

The power transmitted by the system is

$$P = \dot{V}_s \cdot \dot{I}^*_s = \frac{k\sqrt{L_1 L_2}}{\omega_0 L_s L_p} \cdot v_s v_p \sin\theta \tag{18}$$

It can be seen from Equation (18) that the transmission power of the system is mainly affected by $v_s$, $v_p$, and $\theta$. By changing their values, one changes the magnitude and direction of the system power.

Since the system is a high-order circuit, there may be another resonance point near $\omega_0$, and the system characteristics will change abruptly. This can significantly affect the performance of the actual system due to parameter changes. The frequency characteristics of the circuit shown in Figure 6 are given to check whether there is a sharply varying resonance point near $\omega_0$. In order to show the load

effect, the load resistance $R = 8/\pi \cdot R_b$, where $R_b$ is the equivalent resistance of the battery when charging, when the battery charging power is between 5% and 100%, $R$ ranges from approximately 10 to 200 $\Omega$. The frequency characteristics of the input impedance are analyzed at a coupling coefficient of 0.29, as shown in Figure 9. We can see that there is a constant resonant frequency at about 79 kHz. In addition, there are some other resonant frequencies. When the load changes, the lowest and highest resonance points do not change, and they change with the coupling coefficient. It should be noted that near $\omega_0$ there may be another resonance point under certain conditions. However, the change from $\omega_0$ to the adjacent resonance point is smoother, which means that a sudden change does not occur when the operating frequency is slightly shifted from $\omega_0$.

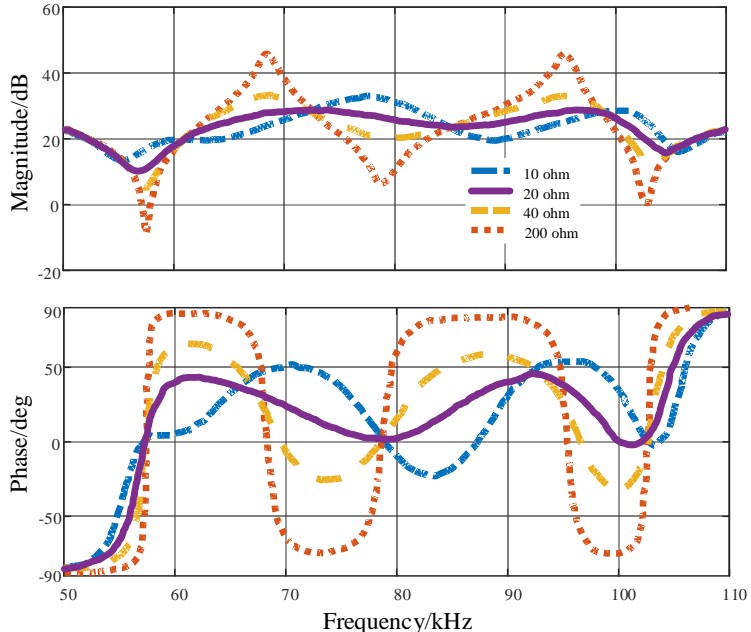

**Figure 9.** Frequency characteristics of the input impedance.

## 5. Simulation Analysis

In order to verify the feasibility and effectiveness of the proposed charging system and control strategy, a simulation was performed in MATLAB/Simulink. Taking into account various factors, the required parameters are shown in Table 3 below. The grid voltage is a three-phase symmetrical voltage with an effective value of 220 V, the battery voltage is 220 V, and $f_T$ is the converter switching frequency.

**Table 3.** Experimental parameters.

| Parameter | Value |
|---|---|
| $L_s$, $L_p$ | 67 μH |
| $C_s$, $C_p$ | 95 nF |
| $C_1$, $C_2$ | 15 nF |
| $L_1$, $L_2$ | 370 μH |
| $k$ | 0.29 |
| $f_T$ | 79 kHz |

Theoretical results of grid voltages and currents over a time frame of 0.08 s are illustrated in Figure 10. Figure 10a is the waveform in the forward direction, and Figure 10b is the waveform in the reverse direction. In the charging mode, the currents and the voltages waveforms basically change in phase, and in the discharging mode, the currents and the voltages waveforms also change in the opposite phase. As a result, high power factors of 0.93 in the charging mode and 0.92 in the

discharging mode were measured. The slight deviation of the power factors from the unit power factor can attributed to the LC low-pass filter and tuning imperfections. In the forward transmission mode, the power can reach 1.53 kW, and it can reach 1.36 kW in the reverse transmission mode. This is because the voltage of the electric vehicle battery is relatively low compared to the grid voltage.

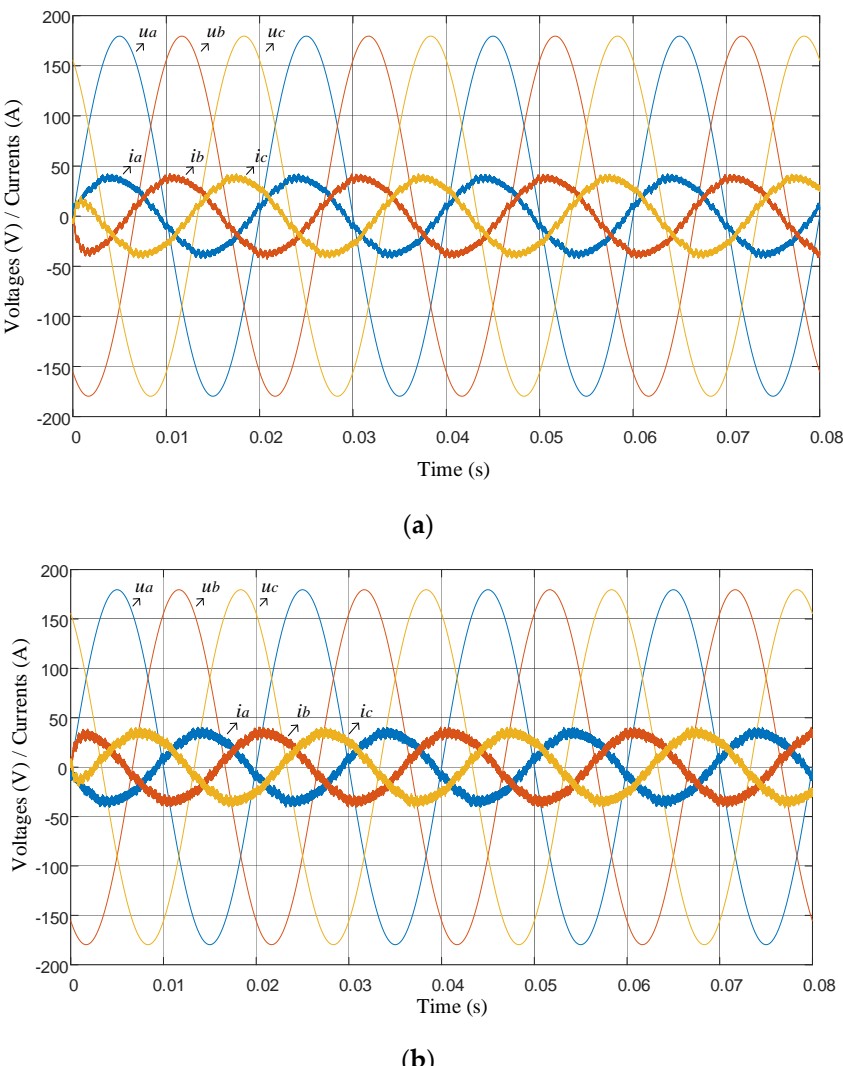

**Figure 10.** Three-phase voltages and currents in (**a**) the forward direction and (**b**) the reverse direction.

In Figure 11, the matrix converter outputs a steady-state voltage and current waveform, which is also the steady-state voltage and current waveform of the resonant tank input. Figure 11a is the waveform in the forward direction, and Figure 11b is the waveform in the reverse direction. As can be seen from the figure, the converter realizes the conversion from low frequency to high frequency. At the same time, it can be seen that the voltage transition is always when the current is 0; that is, the on/off of the switch is performed when the current is 0, indicating that the matrix converter realizes the soft switching operation in the full range during steady state operation. It can also be seen from the figure that the current follows the change of voltage, the frequency and phase are consistent, and the converter has a high power factor.

Figure 12 shows the steady-state voltage and current waveforms of the resonant loop output, and those are also the steady-state input voltage and current waveforms of the full-bridge converter. Figure 12a is the waveform in the forward direction, and Figure 12b is the waveform in the reverse direction. It can be seen that the wireless power transfer enables high power factor operation through compensation of the primary and secondary bilateral LCCL resonant tank. Full-bridge converter also

realizes full range soft switching operation. At the same time, it can be seen that the bilateral LCCL resonant circuit has the ability to achieve constant voltage and constant current output.

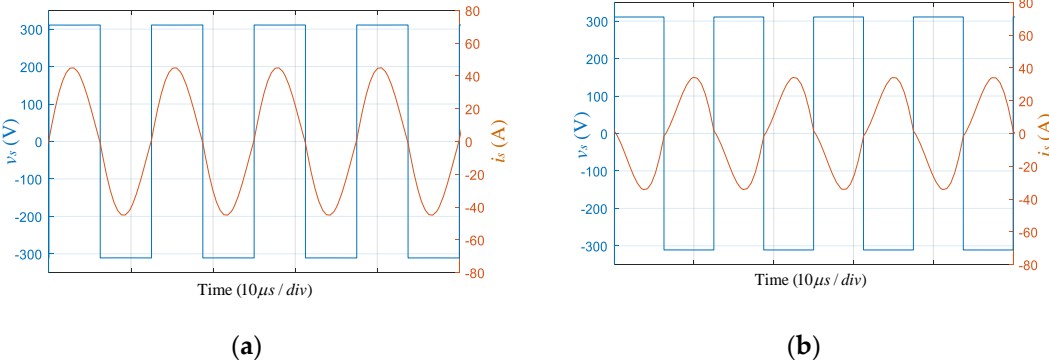

**Figure 11.** $v_s$ and $i_s$ in (**a**) the forward direction and (**b**) the reverse direction.

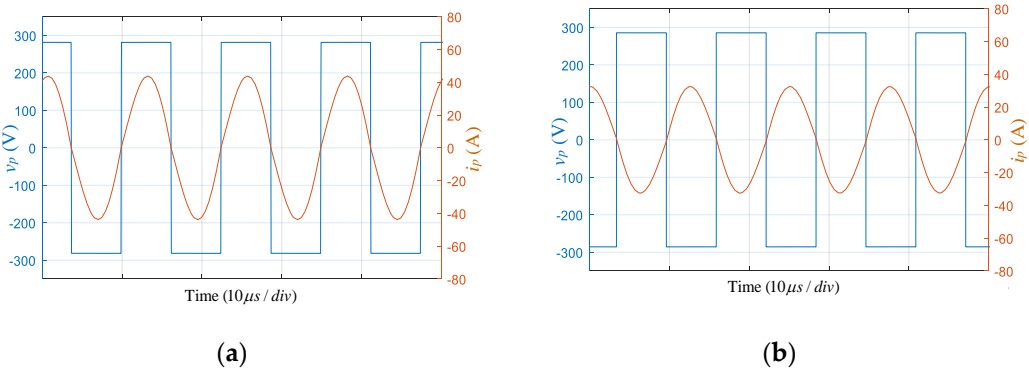

**Figure 12.** $v_p$ and $i_p$ in (**a**) the forward direction and (**b**) the reverse direction.

Figure 13 shows the current waveforms output by the electric vehicle; Figure 13a is the waveform in the forward direction, and Figure 13b is the waveform in the reverse direction. In the forward direction, the current is negative, indicating that the electric car is charging at this time. In the reverse direction, the current is positive, indicating that the electric vehicle is discharging at this time. It can prove the feasibility of the system proposed in this paper and realize the wireless charging and discharging of electric vehicles.

Figure 14 illustrates the THD of the grid current $i_a$ up to the 50th harmonic of the prototype for forward and reverse power operation against the $\theta$, which again ranges from $6/\pi$ to $\pi/2$ in the forward direction and from $-6/\pi$ to $-\pi/2$ in the reverse direction. The overall grid current's harmonic content for reverse power operation is marginally higher than that of the forward power operation; hence the THD for forward power operation is slightly lower than that of the reverse power operation, as depicted in Figure 14. The THD peaks at 4.8% and 5.1% for forward and reverse power operation, respectively, at $|\theta| = \pi/6$ and then maintains a steady lower trend with increasing $|\theta|$ values.

The efficiency of the system, from the grid to the dc battery for electric vehicle, is shown in Figure 15 for both directions of power flow at different values of $\theta$. As evident from Figure 15, the efficiency profile of the system for the forward power flow is similar to that of the reverse power flow in general, with some minor differences. The system exhibits its highest efficiency of 92.6% at $\theta = \pi/2$ for the forward power operation, whereas the highest efficiency for the reverse power operation is 92.3% at $\theta = -\pi/2$. Furthermore, it is observed that the efficiency decreases with decreasing $|\theta|$ values.

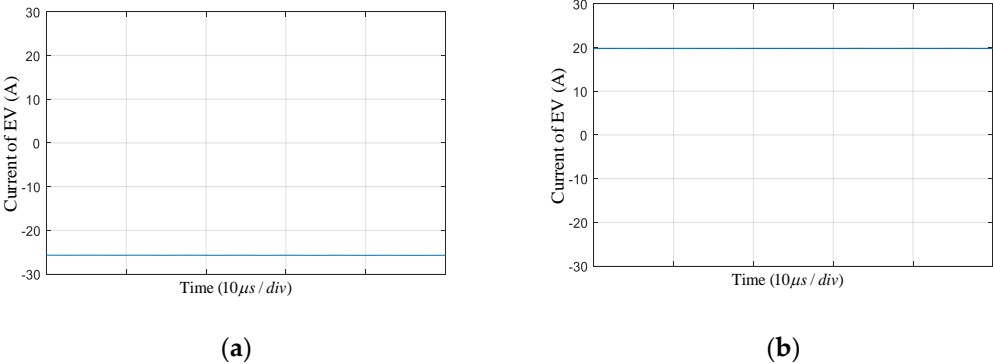

**Figure 13.** Current of EV in (**a**) the forward direction and (**b**) the reverse direction.

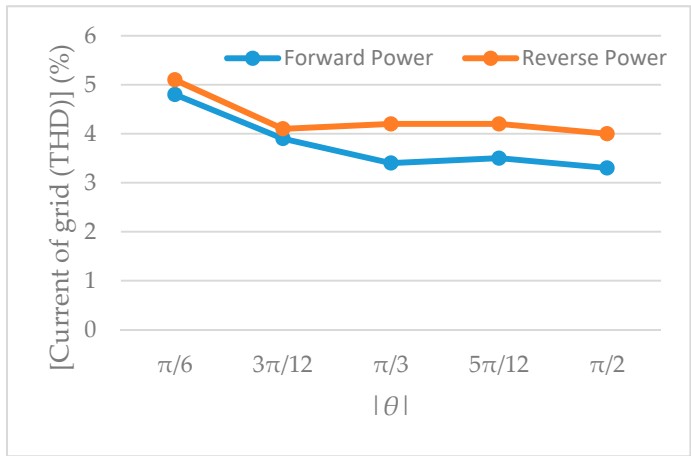

**Figure 14.** THD of grid current $i_a$.

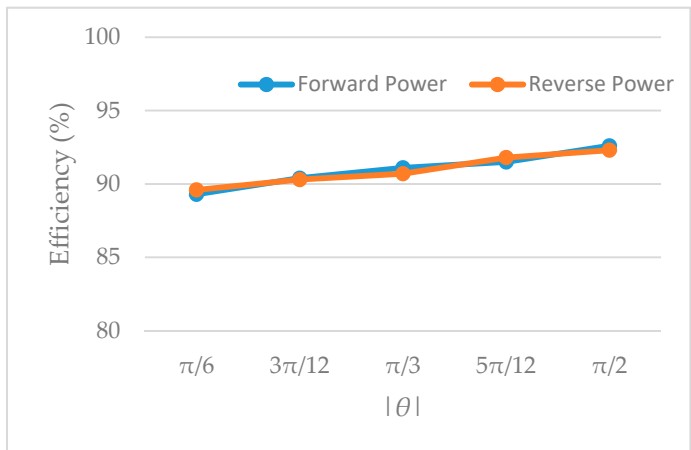

**Figure 15.** Efficiency of the system.

## 6. Conclusions

In this paper, a new type of electric vehicle wireless power transfer system based on a direct three-phase AC–AC matrix converter is proposed, and a new, simple control strategy is proposed for this system. Compared with the traditional space vector control, the number of switches is reduced, the soft switching operation is realized in the full operation range, the switching loss is reduced, and the system power factor is improved. The system does not need any intermediate energy storage components to realize the conversion of low-frequency alternating current to high-frequency

alternating current. Combined with the bilateral LCCL resonant compensation loop proposed in this paper, through theoretical analysis and experimental verification, the system has constant voltage and current output characteristics, and realizes the wireless charging and discharging of electric vehicles.

**Author Contributions:** R.W. conceived the theory; M.H. performed the experiments; C.L. and W.W. contributed materials; R.W. and M.H. wrote the paper. M.H. and W.W. reviewed and edited the manuscript. All authors have read and agreed to the published version of the manuscript.

**Funding:** This research is funded by National Key R&D Program of China under grant number 2017YFB0903300.

**Conflicts of Interest:** The authors declare no conflict of interest.

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
