# Peer review of "A Direct Three-Phase AC–AC Matrix Converter-Based Wireless Power Transfer System for Electric Vehicles"

_applsci, doi:10.3390/app10072217_

Round 1
Reviewer 1 Report
The authors made the suggested modifications. Therefore, the paper can be acpeted.
Minor comment:
In fig 10, even if I understand which are the current and voltage waveforms, the colours used to distinguish voltages and currents are the same. Please change the colour or the type lines.
Author Response
Point: In fig 10, even if I understand which are the current and voltage waveforms, the colours used to distinguish voltages and currents are the same. Please change the colour or the type lines.
Response: Thank you for your affirmation of our work and pointed out the problem in the manuscript. In response to this problem, we have modified the reference forms of currents and voltages in Figure 10 (page 10, lines 274-278) to correspond to their waveforms, as shown in the upload PDF file.

Reviewer 2 Report
This reviewer would like to thank the authors for their effort in replying all the comments and questions. The additions made in the revised manuscript are enough to improve the overall content. Just a minor detail regarding the new paragraph added in page 11, line 315: "The PHD peaks ...." should be "the THD peaks..." instead. Besides this typo, there is no additional comments addressed to the authors.Author Response
Point: Just a minor detail regarding the new paragraph added in page 11, line 315: "The PHD peaks ...." should be "the THD peaks..." instead. Besides this typo, there is no additional comments addressed to the authors.
Response: Thank you for affirming our work and pointing out the problem in the manuscript. It is not appropriate to make mistake in the word spelling again. Please forgive us for the mistake and thank you again for your comments. In the new manuscript, we have modified the “PHD” to “THD” according to your suggestions in page 11, line 315.
This manuscript is a resubmission of an earlier submission. The following is a list of the peer review reports and author responses from that submission.
Round 1
Reviewer 1 Report
The work is interesting and clear. I understood that the de-coupling method means to threat the matrix converter as an indirect matrix converter. Of course this is a simple control strategy. Nevertheless, in the reported analysis nothing is said about the waveform and power factor of the current of the main ac grid. I think it is interesting to analyse such currents and their harmonic content. You can refer to [1] where is shown an analysis of a three-phase to three-phase matrix convert under different conditions.
Minor comments:
In line 28 it is better use the word electric vehicles instead of automobiles.
In line 118 the title is Matrix Concerter instead of converter
There is a problem in the superscript. The symbol “&” appears slightly hiding the other letter.
[1] Simone Barcellona, Maria Stefania Carmeli, Gabrio Superti-Furga, “Comprehensive harmonic analysis of matrix converter under unbalanced/distorted conditions”, Electric Power Systems Research, Vol. 96, 2013, pp. 296-310, doi.org/10.1016/j.epsr.2012.11.017.
Reviewer 2 Report
This paper proposes a wireless charging system based on a direct matrix converter (MxC) and a LCCL resonant tank. Overall, the use of a MxC as a wireless charging system is not new (take as an example [R1] or [R2]). On top of that, the use of an LCCL tank for compensating the resonant frequency neither is new [R3].
However, the analysis carried out by the authors of this resonant tank and the proposed control strategy based on de-re-coupling for the MxC can be considered as an incremental contribution in this particular field.
In spite of the lack of novelty, the paper is well written and presented. Figures are clear and references to previous works are good. The only minor formatting mistake is related with Fig. 6 (an ampersand appears on top of the voltages and currents symbols). Authors should carefully revise this typo that is repeated on equations (8) to (18).
This reviewer suggests to the authors to try to enhance the simulation results section (Section 5). Please, consider to include some efficiency estimation about the proposal as well as some THDi and PF measurements for both power flow directions. Ideally, an experimental validation of the proposed converter should add a strong point to the paper.
[R1] Y. Hayashi, H. Motoyama and T. Takeshita, "Wireless Power Transfer System Using Three-phase to Single-phase Matrix Converter," 2018 International Power Electronics Conference (IPEC-Niigata 2018 -ECCE Asia), Niigata, 2018, pp. 356-362.
[R2] J. Noeren, N. Parspour and B. Sekulic, "A Direct Matrix Converter with Space Vector Modulation for Contactless Energy Transfer Systems," 2018 IEEE 18th International Power Electronics and Motion Control Conference (PEMC), Budapest, 2018, pp. 22-27.
[R3] L. Guan, Z. Wang, P. Liu and J. Wu, "A Three-Phase to Single-Phase Matrix Converter for Bidirectional Wireless Power Transfer System," IECON 2019 - 45th Annual Conference of the IEEE Industrial Electronics Society, Lisbon, Portugal, 2019, pp. 4451-4456.